# Reduction in SARS-CoV-2 Virus Infectivity in Human and Hamster Feces

**DOI:** 10.3390/v14081777

**Published:** 2022-08-15

**Authors:** Sébastien Wurtzer, Sandra Lacote, Severine Murri, Philippe Marianneau, Elodie Monchatre-Leroy, Mickaël Boni, Olivier Ferraris, Yvon Maday, Ousmane Kébé, Ndongo Dia, Christophe Peyrefitte, Harry Sokol, Laurent Moulin, Vincent Maréchal

**Affiliations:** 1Research and Development Department, Eau de Paris, 33 Avenue Jean Jaurès, 94200 Ivry-sur-Seine, France; 2ANSES—Laboratoire de Lyon, Unité Virologie, 69007 Lyon, France; 3Nancy Laboratory for Rabies and Wildlife, ANSES, 54220 Malzéville, France; 4French Armed Forces Biomedical Research Institute, 91220 Brétigny-sur-Orge, France; 5Laboratoire Jacques-Louis Lions (LJLL), CNRS, Sorbonne Université, Université de Paris, 75005 Paris, France; 6Institut Universitaire de France, 75005 Paris, France; 7Institut Pasteur de Dakar, Dakar 12900, Senegal; 8Institut Pasteur de la Guyane, 97300 Cayenne, France; 9INSERM, Centre de Recherche Saint-Antoine, CRSA, AP-HP, Saint Antoine Hospital, Gastroenterology Department, Sorbonne Université, 75571 Paris, France; 10Paris Centre for Microbiome Medicine (PaCeMM) FHU, 75571 Paris, France; 11INRAe, UMR1319 Micalis and AgroParisTech, 78350 Jouy en Josas, France; 12INSERM U938, Centre de Recherche Saint-Antoine, Sorbonne Université, 75012 Paris, France

**Keywords:** SARS-CoV-2, transmission, feces, stool, persistence, infectivity

## Abstract

Objective: There is extensive evidence that SARS-CoV-2 replicates in the gastrointestinal tract. However, the infectivity of virions in feces is poorly documented. Although the primary mode of transmission is airborne, the risk of transmission from contaminated feces remains to be assessed. Design: The persistence of SARS-CoV-2 (infectivity and RNA) in human and animal feces was evaluated by virus isolation on cell culture and RT-qPCR, respectively. The exposure of golden Syrian hamsters to experimentally contaminated feces through intranasal inoculation has also been tested to assess the fecal-oral transmission route. Results: For periods that are compatible with average intestinal transit, the SARS-CoV-2 genome was noticeably stable in human and animal feces, contrary to the virus infectivity that was reduced in a time- and temperature-dependent manner. In human stools, this reduction was variable depending on the donors. Viral RNA was excreted in the feces of infected hamsters, but exposure of naïve hamsters to feces of infected animals did not lead to any productive infection. Conversely, hamsters could be experimentally infected following exposure to spiked fresh feces. Conclusion: Infection following exposure to naturally contaminated feces has been suspected but has not been established so far. The present work demonstrates that SARS-CoV-2 rapidly lost infectivity in spiked or naturally infected feces. Although the possibility of persistent viral particles in human or animal feces cannot be fully ruled out, SARS-CoV-2 transmission after exposure to contaminated feces is unlikely.

## 1. Introduction

SARS-CoV-2, the causative agent of COronaVIrus Disease 2019 (COVID-19), belongs to the *Coronaviridae*, a large family of enveloped single-stranded positive-sense RNA viruses. The life cycle of the virus takes place mainly in the upper and lower respiratory tracts. SARS-CoV-2 spreads from person to person mainly through the transmission of contaminated droplets or aerosols produced from the respiratory tract of infected individuals [1]. The common clinical symptoms for COVID-19 are fever, dry cough, dyspnea, muscle soreness, headache, anosmia, and great tiredness. Additionally, two meta-analyses of data reported that 17.6% and 12% of patients with COVID-19 experienced gastrointestinal (GI) symptoms, respectively [2,3]. Diarrhea (7.7%), nausea or vomiting (7.8%) and abdominal pain are the more prevalent GI symptoms [4]. The variation in the proportion of patients with GI disorders among different studies might be related to the inclusion of other non-specific symptoms of GI and the course of symptom reporting (admission/hospitalization), especially in the case of medication-induced disorders. It is suggested that GI symptoms might be a sign of COVID-19 onset in some patients [5]. As with many other respiratory viruses [6], the SARS-CoV-2 RNA genome has been detected in infected patient fecal samples [2,3]. The presence of the virus genome in feces does not appear to be strictly correlated with the presence of GI symptoms [7] or with COVID-19 severity [8]. The detection of the viral genome might result from swallowing respiratory secretions, but strong evidence supports an active replication in intestinal cells. Indeed, absorptive enterocytes from the ileum and colon were found to express high levels of the cellular surface protein ACE2 and a serine protease TMPRSS2, which are used by SARS-CoV-2 for the binding of its spike proteins to the cell and for the priming of the spike, respectively [9]. The combination of high concentrations of viral RNA [10], the detection of subgenomic RNA [11] and viral antigens [12] as well as prolonged detectability in fecal samples compared with respiratory samples [10,13,14,15,16] supports an active replication of SARS-CoV-2 in the GI tract. Finally, viral lesions were observed in the GI tract of patients [17,18].

The presence of virus RNA in the stool provides at least two main implications. First, numerous studies have demonstrated the value of exploiting viral RNA excretion in feces for wastewater-based epidemiology purposes [19,20,21,22]. The quantification of viral genomes in raw sewage allows for the dynamic estimation of virus circulation in populations without the possible biases linked to the patient-centered screening strategy and to the important proportion of asymptomatic cases. A better knowledge of the viral genome stability and the viral excretion kinetics in the stools of infected persons, especially according to the circulating variants or the vaccination status, is necessary to refine this approach. Moreover, the possibility of a modification of cellular tropism shift for new variants may likely impact viral shedding especially in stool [23].

Secondly, the infectivity of viral particles in the stools of infected persons may be responsible for fecal-oral contamination by contaminated surface or aerosols, or through their elimination in wastewater, or by fecal microbiota transplantation (FMT) [24,25]. Several studies have shown that raw and treated wastewater is unfavorable for the persistence of the virus SARS-CoV-2 under an infectious form [26,27] in accordance with the usual claim that enveloped viruses do not survive long in a complex water environment. Only a few publications have reported the presence of infectious [28,29,30,31] or intact viral particles [32] in human feces, or potential transmission events with a fecal origin. SARS-CoV-1 transmission was linked to fecal aerosols during the 2003 outbreak that affected more than 300 residents of Amoy Gardens [33]. No similar outbreak has been reported to date for SARS-CoV-2, suggesting that aerosols containing fecal virus may be responsible for its transmission [34]. McDermott and colleagues examined whether fecal bioaerosols are a route of transmission for SARS-CoV-2 in hospitals [35]. Other investigations by Jun Yuan and colleagues also support the possible transmission of SARS-CoV-2 via a defective toilet sewage pipe [36]. These observations were compatible with the detection of aerosols with high virus concentrations inside a patient mobile toilet room at Fangcang Hospital [37]. To date, no case of SARS-CoV-2 transmission through FMT has been reported.

In addition to in vitro models supporting SARS-CoV-2 intestinal infection [38,39,40], various animal models have been proposed for studying SARS-CoV-2 infection. Some of them report evidence of intestinal SARS-CoV-2 infection. The golden Syrian hamster (*Mesocricetus auratus*) is an especially suitable animal model presenting continuous viral RNA shedding in feces after infection through fomites [41,42,43]. Furthermore, infectious particles were isolated from nasal swabs after the intragastric transfer of fecal supernatant in ferret [44,45,46]. Most SARS-CoV-2 variants cannot infect wild-type mice since the virus does not seem to be able to bind to mouse ACE2 [47], but human ACE2 knock-in mice can be infected by intragastric SARS-CoV-2 [48].

Up to now, the evidence of infection events with fecal origin has been poorly documented and raises the need for further studies dealing with the persistence of viral infectivity for understanding the risk of fecal-oral or fecal-nasal infection routes [49].

In the present study, we first investigated viral persistence and infectivity in human stools from different geographic origins and in spiked SARS-CoV-2 feces of hamster. As enteric viruses are mainly responsible for gastroenteritis and are transmitted by the fecal-oral route, a comparison of the persistence of infectivity of SARS-CoV-2 and Coxsackievirus B5 in the stool was done. In order to confirm the results, SARS-CoV-2 was propagated in the golden hamster model. Then, the infectivity of feces from experimentally infected animals or from virus-spiked fecal samples was assessed.

## 2. Material and Methods

### 2.1. In Vitro Assay/Virus Stock Preparation

Coxsackievirus B5 (CV-B5, #ATCC VR-185) was cultivated on confluent monolayer cultures of Buffalo Green Monkey kidney (BGMK, #ATCC PTA-4594) cells. Cells were grown in Dulbecco’s Modified Eagle’s Medium (DMEM) high glucose (Dutscher, Bernolsheim, France, #L0103) supplemented with 2% fetal bovine serum (PanBiotech, #3301-P113103), non-essential amino acids (Dutscher, µFrance #X0557). The supernatant was clarified by centrifugation at 2000× *g* for 15 min, then ultracentrifuged at 150,000× *g* at 4 °C for 2 h through a 40% sucrose cushion. The pellet was resuspended in phosphate-buffered saline (PBS) 1× pH 7.4. Further purification was performed by ultracentrifugation on cesium chloride gradient (from 1.2 g/L to 1.5 g/L) at 100,000× *g* for 18 h. The fraction containing the viruses was desalted using Vivaspin 20 ultrafiltration units (10 kDa MWCO) (Sartorius, Göttingen, Germany, #VS2001). Viruses were stored at −80 °C before utilization.

SARS-CoV-2 20/0001 (BetaCoV/France/IDF0372/2020/SARS-CoV-2 collected in January 2020 and kindly given by Institut Pasteur, France), was cultivated on confluent monolayer cultures of VERO E6 cells (ATCC CRL-1586). Cells were grown in Dulbecco’s Modified Eagle’s Medium GlutaMAX (Gibco, New York, NY, USA, #31966047), TPCK trypsin (1 µg/mL) (Gibco, New York, NY, USA, #20233) without fetal bovine serum. The supernatant, collected after the observation of a cytopathic effect, was clarified by centrifugation at 2000× *g* for 15 min and stored at −80 °C before utilization.

### 2.2. Human Fecal Samples Collection

French fecal samples were obtained from healthy subjects (men and women) of the Suivitheque biobank (Paris, France). Approval for human studies was obtained from the local ethics committee (Comité de Protection des Personnes Ile-de-France IV, IRB 00003835 Suivitheque study; registration number 2012/05NICB). African fecal samples (men and women) were collected from the Syndromic Surveillance System implemented in Senegal conducted by the Institut Pasteur de Dakar. Approval for human studies was obtained from the national committee (registration number 00029/MSAS/DPRS/CNERS).

### 2.3. Detection of SARS-CoV-2 in Spiked Assays

Fresh stool samples were collected in non-symptomatic patients (five in Paris, France and nine in Dakar, Senegal). The stools were negative for SARS-CoV-2 and enterovirus genomes. Ten grams of stool sample was diluted in 20 mL of complete cultivation medium and centrifuged at 3000× *g* for 10 min to remove the largest particles, and supernatants were filtered on low binding protein membrane with 0.45 µm porosity. The filtrates were stored at 37 °C and used within the following 24 h.

CV-B5 or SARS-CoV-2 were spiked in the filtered samples. Virus titration was immediately performed after the spiking, then during the experimentation after incubation steps at 4 °C or 37 °C for 30 min or 6 h. As a control, DMEM was used in the spiking experiments. Virus infectivity and viral RNA detection were assessed by endpoint plaque assay and RT-qPCR, respectively.

### 2.4. Infectious Virus Quantification by Endpoint Plaque Assay

The viral titer of the samples was evaluated for half by endpoint assay and for the other half by RT-qPCR. Half of the spiked samples were titrated (CV-B5 and SARS-CoV-2) by standard 10-fold dilutions in 96-well plates on VERO E6 cells (10^5^ cells per well), with twelve replicates per dilution. After a 6-day incubation step, cytopathic effects were observed, and positive wells were counted. Viral titer was estimated using the Spearman-Kärber method. The results are expressed as 50% tissue culture infective dose (TCID50) per ml.

### 2.5. Viral RNA Detection by RT-qPCR

The other half of the spiked samples were lysed by adding two volumes of TRIZOL (Lifetechnologies, Illkirch-Graffenstaden, France #15596018) and extracted using a Qiasymphony PowerFecal Pro (QIAGEN, Les Ulis, France #938036) kit on a QIAsymphony automated extractor (QIAGEN) according to the manufacturer’s protocol. Extracted nucleic acids were filtered through a OneStep PCR inhibitor removal kit (Zymoresearch, Freiburg im Breisgau, Germany, #ZD6030).

The RT-qPCR primers and PCR conditions used herein have been previously described [20]. Briefly, the amplification was done using Fast virus 1-step Master mix 4×(Lifetechnologies, France #4444434). Detection and quantification were carried on the gene E by RT-qPCR. Positive results were confirmed by amplification of viral RNA-dependent RNA polymerase (RdRp) gene. The limit of detection (95% probability of detection) was estimated to be around 10 genome units per reaction for each amplification. An internal positive control (IPC) was added to evaluate the presence of residual inhibitors. The IPC consists of a plasmid containing beta-actin gene flanked by enterovirus-specific primers [50].

The quantification was performed using a standard curve based on full-length amplicon cloned into pCR2.1 plasmid (Invitrogen, #452640). Amplification reaction and fluorescence detection were performed on Viia7 Real Time PCR system (Lifetechnologies).

### 2.6. Animal Experimental Assay

All animal experiments were carried out in a biosafety level 3 (BSL3) facility in the Plateforme d’Expérimentation Animale (ANSES—laboratoire de Lyon, Lyon, France). The experiments (summarized in Figure 1) were approved by the Anses/ENVA/UPEC ethics committee and the French Ministry of Research (Apafis n°24818-2020032710416319).

### 2.7. Feces Infectivity

All the experiments were performed with 8-week-old female golden Syrian hamsters (*Mesocricetus auratus*, strain RjHan:AURA—Janvier Labs, Le Genest, St Isle, France). Three animals, called “donor hamsters” (Figure 1B), were inoculated intranasally either with 10^4^ plaque-forming units (PFU) of SARS-CoV-2 (Wuhan strain UCN1, kindly provided by University Hospital of Caen, Caen, France) (*n* = 2) or DMEM for mock animals (*n* = 1) in a volume of 40 µL (20 µL in each nostril).

Mock and SARS-CoV-2-infected hamsters were euthanized under anesthesia at day 4 post infection. Feces were collected, weighted, and homogenized in DMEM with stainless steel beads (QIAGEN, France # 69989) for 3 min using TissueLyserII (QIAGEN, # 85300). Feces homogenates were clarified by centrifugation (3000× *g* at 4 °C for 10 min), filtered on 0.45 µm and immediately used for inoculation of “recipient hamsters”. Three different inocula were prepared: negative and positive samples (filtered supernatant from, respectively, mock healthy and infected hamster feces), and spiked feces corresponding to negative feces in which 10^4^ PFU of SARS-CoV-2 were added. Before inoculation, feces were crushed, centrifuged (3000 rpm, 10 min), and filtered. All these steps could last 30 min maximum as indicated in Figure 1 by “intranasal inoculation within 30 min”. A positive control (10^4^ PFU of SARS-CoV-2) was also used to compare the infection pattern (infected organs, clinical signs) between different inocula. It ensured the efficiency of the virus to cause an infection.

Lungs were harvested from “donor animals” to check the efficiency of the infection. All tissue samples from infected and mock animals were stored at −80 °C until virus titration.

Groups of “recipient hamsters” were inoculated with 40 µL of either negative feces or positive feces, spiked feces, or positive control. Four days later, hamsters were euthanized. Lungs and feces were collected, weighted, homogenized, and clarified as described before.

### 2.8. Virus RNA Detection in Feces Samples

Feces from two non-infected hamsters were collected and weighted in order to obtain 2 pools of feces with similar weight. Samples were then homogenized in DMEM. Both crude homogenates were spiked with 2 × 10^4^ PFU of SARS-CoV-2 per 100 µL. After centrifugation at 3000× *g* for 10 min, supernatants were filtered on 0.45 µm membrane.

Two time conditions (30 min and 6 h) and three temperature conditions (4 °C, 25 °C, and 37 °C) were tested to control RNA persistence over time for each pool of feces. Titrations were immediately performed following the sampling step. Viral RNA was extracted from 100 µL of tissue supernatants using a QIAamp Viral RNA kit (QIAGEN, France, # 52904) following the manufacturer’s instructions except for Lysis buffer (AVL), in which 2.5% of triton X100 was added. The eluted RNA was stored at −80 °C.

### 2.9. SARS-CoV-2 Genome Detection

RT-qPCR was performed with the SuperScript^TM^ III Platinum^TM^ One-Step qRT-PCR kit (Invitrogen, France # 11732020) and E gene set of oligonucleotides as described by Corman [51], using a LightCycler 480 instrument (Roche, Meylan, France).

### 2.10. Titration

Viral titers were determined by endpoint plaque assay on Vero E6 cells. Samples were diluted by standard 10-fold dilutions prior to the infection of cells. One hour later, plaque assays were then overlaid with carboxy-methylcellulose mix (CMC 3.2% DMEM 5% FBS (*v*/*v*)) and after 5 days were fixed and stained with a crystal violet solution (3.7% formaldehyde (*v*/*v*), 0.2% crystal violet(g/V)).

### 2.11. Evaluation of Stool-Induced Cytotoxicity

VERO E6 cells were incubated with raw or diluted non-spiked stool filtrates for up to 24 h, then washed 2 times in PBS 1×. Cell viability was estimated using live/dead labeling and counting by Flow cytometry on Cube 8 instrument (Sysmex, Villepinte, France). Live cells were SYTO 9 positive and Propidium iodide negative (data not shown).

### 2.12. Statistical Analysis and Plots

All statistical analysis and plots were done using GraphPad Prism 9.0 software. For comparison based on spiked samples (Figure 2), the quantifications were compared between the different conditions using Kruskal–Wallis test and Dunn’s multiple comparisons test (*n* = 18 for French stools and *n* = 30 for African stools).

## 3. Results

### 3.1. SARS-CoV-2 Infectivity in Human Feces

Although SARS-CoV-2 was frequently detected in human stools by RT-qPCR at high concentrations, there are only a few reports of positive virus isolation from stools in cell culture, suggesting that the virus may be poorly infectious in feces. To evaluate SARS-CoV-2 infectivity in human feces, five stools from COVID-19-negative healthy adult donors, exhibiting a Caucasian profile, were spiked with infectious virus and incubated for 30 min at room temperature or 6 h either at 4 °C or at 37 °C (Figure 1A). This incubation time was set at 6 h, since it was estimated to correspond to an average human intestinal transit time, independently of age [52]. Then, viral genomes were quantified by RT-qPCR and the viral infectivity was evaluated by endpoint plaque assay titration. Coxsackievirus B5, a non-enveloped RNA virus commonly involved in gastroenteritis, was used as a positive control. No cytotoxicity was observed in the endpoint plaque assay (data not shown). RNA genome detection of both Coxsackievirus B5 (Figure 2A) and SARS-CoV-2 (Figure 2B) was not affected in these experiments. Just after feces spiking (30 min at 4 °C), less than 1-log reduction of infectivity was observed compared to SARS-CoV-2 incubation in DMEM (Figure 2D). The almost immediate titer reduction, i.e., after a 30 min-incubation at 4 °C, was not significantly impacted by any longer incubation times (up to 6 h) at 4 °C. Conversely, an important reduction in SARS-CoV-2 infectious titer was observed in most samples when the spiked stools were incubated at 37 °C (Figure 2D). As expected, Coxsackievirus B5 infectious titer was not significantly modified in the same conditions (Figure 2C).

Similar experiments were conducted with feces from nine SARS-CoV-2 negative African donors (Figure 1A). The results confirmed that SARS-CoV-2 RNA genome titer was not significantly modified after an incubation for 6 h at 37 °C (Figure 2E), whereas the infectious viral titer evaluated was dramatically reduced in the same conditions when compared to the incubation in DMEM (Figure 2F).

Altogether, these results suggested that human stools induced a high reduction in SARS-CoV-2 infectivity after incubation at 37 °C for 6 h but that did not significantly affect the viral RNA load. As expected, no significant decrease was observed for Coxsackievirus B5. These results based on human feces described both a time- and temperature-dependent decrease as well. However, some variation was observed between donors, as residual infectivity was detected in the feces of some donors regardless of origin.

**Figure 2 viruses-14-01777-f002:**
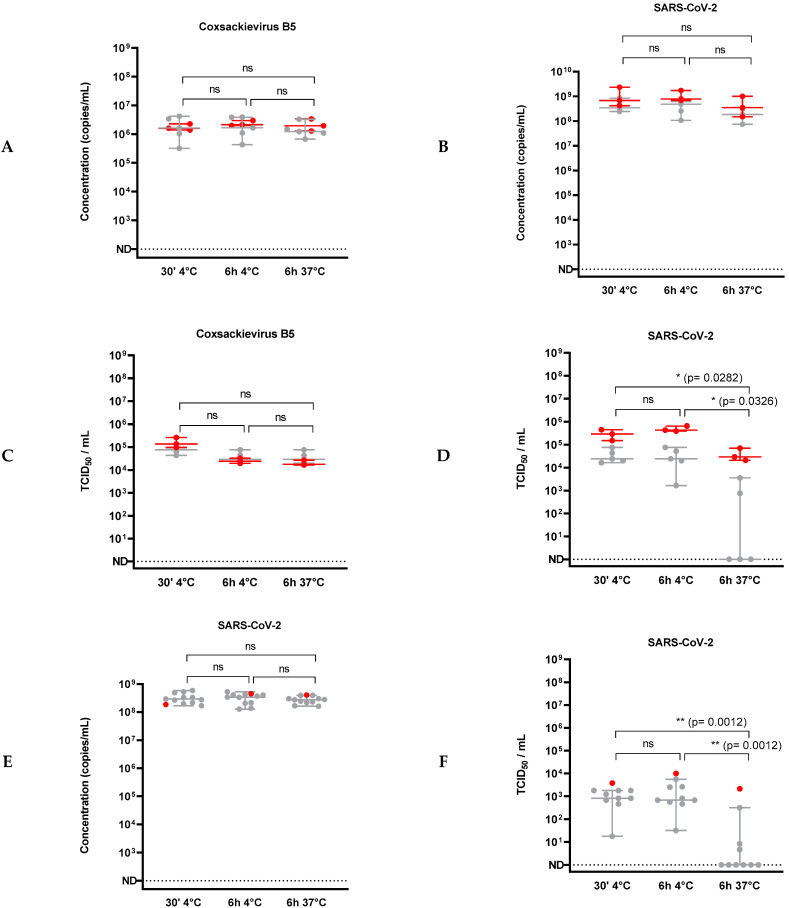
Persistence of Coxsackievirus B5 and SARS-CoV-2 in human feces (**A**–**F**). Fresh human feces (grey) were inoculated with infectious viral suspension compared to cell culture medium (red) for up to 6 h. Titration of Coxsackievirus B5 (panel (**C**)) or SARS-CoV-2 (panel (**D**)) was done by endpoint plaque assay. Infectious Coxsackievirus B5 concentration did not show any significant change in spiked feces samples. Persistence of viral RNA (panel (**E**)) or infectious particles (panel (**F**)) of SARS-CoV-2 in feces from African negative donor was analyzed. Statistical tests were considered as positive if *p*-value < 0.05. *: *p*-value < 0.05; **: *p*-value < 0.01. ns = not significant.

### 3.2. SARS-CoV-2 Infectivity in Golden Hamster Feces

We next designed an exploratory approach to investigate the possibility of transmission from contaminated feces in a golden hamster model. Two hamsters (*n* = 2) were experimentally infected with 10^4^ PFU and were euthanized 4 days post infection (Figure 1B). Previous work by our group showed that this procedure was efficient to generate a high viral content in nasal turbinates, trachea, lungs saliva, and feces [42]. As expected, the SARS-CoV-2 genome was detected in the lung and feces from donor specimen (Figure 3A), but no infectious SARS-CoV-2 could be isolated from hamster feces neither in cell culture (Figure 3A) nor after intra-nasal inoculation to new recipient specimen (*n* = 8) despite a genomic viral load of 4 × 10^6^ and 6 × 10^7^ genome copies/hamster (Figure 3B).

In order to evaluate the persistence of infectious virus in hamster feces, fresh feces from healthy hamsters were spiked with about 10^4^ PFU of SARS-CoV-2 (Figure 1C). Spiked feces were inoculated to recipient hamster specimens (*n* = 4) or incubated for 6 h at 4 °C, 25 °C, or 37 °C before inoculation to cell culture. All recipient hamsters were challenged, and concentrations of viral genome were detected in various organs (nasal turbinates, trachea, lungs saliva, and feces) similarly to the former specimen infected with infectious virus (Figure 3B). The spiked hamster feces showed stable viral RNA load measured by RT-qPCR in all tested conditions (Figure 4A). Conversely, the measurement of residual infectious particles demonstrated that hamster feces induced a marked and rapid decrease in infectivity. The infectivity decreased with time (from 30 min up to 6 h) and with an increasing incubation temperature (ranging from 4 °C to 37 °C) (Figure 4B) as previously observed in human stool samples. Altogether, these results indicate that hamster feces markedly reduced SARS-CoV-2 infectivity in a time- and temperature-dependent manner without significantly affecting the amount of the detected viral genome. Nevertheless, a sufficient number of infectious virions persisted in order to infect a new specimen. In a global approach, by combining the inoculum of infectious virus spiked in negative stools (10^4^ PFU) and the reduction of the infectious titer (mean value −2.9 log as shown on Figure 4B) observed after 30 min at 4 °C (time needed to process the feces sample for inoculation), about 13 (10^4−2.9^) PFU/animals were sufficient to induce a replicative infection in golden hamsters after an intranasal inoculation.

## 4. Discussion

Contaminated respiratory secretions (droplets and aerosols) are considered to be the main mode of human-to-human transmission of SARS-CoV-2. However, various elements suggest that fecal-oral contamination may be an alternative route of transmission for SARS-CoV-2 in humans. First of all, evidence of intestinal replication has been demonstrated: clinical evidence with the report of numerous infected patients suffering from gastroenteritis symptoms [2,3], significant fecal excretion of viral genomes in feces [10], and histological analyses revealing the presence of virions and infiltration of pro-inflammatory cells in the duodenum and rectum [12,53]. In addition, viral replication has been demonstrated in human intestinal cell lines and intestinal organoids [39,54], as well as animal models [42]. A viral intestinal reservoir could be implicated in long-COVID [55]. The description of a fecal-oral infection with SARS-CoV in 2003 supported such a possibility [33]. Despite this evidence, there are very few case reports of proven contamination through SARS-CoV-2 fecal shedding [34,35,36,37].

To establish transmission from exposure to contaminated feces, SARS-CoV-2 must be able to remain infectious in feces and to initiate replication in respiratory tract or gut eventually. Unlike enteric viruses, SARS-CoV-2 has a relatively fragile lipid membrane. SARS-CoV-2 is inactivated by simulated human colonic fluid within 10 min [54]. However, as with influenza viruses, mucosal secretions or food intake may protect SARS-CoV-2 from such inactivation [56]. Viral persistence on food and non-food contact surfaces was intensively described early in the pandemic [57], and fecal shedding of viral genomes was widely reported and used particularly for epidemiological purposes [58].

In this study, the fecal-nasal transmission was investigated in an animal model after intranasal inoculation of fecal samples contaminated with SARS-CoV-2 RNA [42]. Intranasal instillation of a sufficient quantity of virions has initiated infection and resulted in pleiotropic viral involvement with a maximum viral load observed in the lungs of the animals. Viruses isolated from the lungs were infectious, whereas those extracted from the feces could not replicate in culture. Inhalation of fecal extracts from infected hamsters did not transmit infection to new animals despite high genomic viral input. These results confirmed previously published results in the same animal model [42]. These results indicated that the feces of infected animals did not contain replicative virus or below minimal infectious dose of this model. The minimum infectious viral dose for SARS-CoV-2 is poorly documented. The lowest dose was 10^3^ PFU in hamsters [42] and 10^2^ PFU in transgenic K18-hACE2 hamsters [59]. Based on the inoculum of infectious virus spiked in stools and the experimental reduction of the infectious titer in hamster feces, 13 PFU of SARS-CoV-2 were enough to infect a hamster through fecal-nasal transmission.

Spiking fresh uninfected hamster feces with infectious SARS-CoV-2 for 30 min—i.e., the time needed to process the feces sample for inoculation of cell culture—led to the infection of naïve animals. This demonstrated that the virus was not completely inactivated upon short contact time with feces and that a sufficient amount of infectious virus persisted in these conditions. To evaluate the inactivation of SARS-CoV-2 in both hamster and human feces, the contact time and the incubation temperature were tested. Spiking infectious viruses in negative hamster feces showed a temperature- and time-dependent viral inactivation on Vero E6 cells. The effect of temperature on SARS-CoV-2 inactivation has already been reported in wastewater samples [27]. The viral genomic load was not modified, even when complete inactivation of the virus infectivity was observed. Regarding human stools, two different geographical origins were tested and a comparison with Coxsackievirus B5 as a surrogate of enteric virus has been done. While Coxsackievirus B5 infectious virions and viral RNA persisted in feces, SARS-CoV-2 viral RNA was preserved from degradation, but virions lost their infectivity in feces extract after 6 h at 37 °C. The same reduction was observed independently of the geographical origin of the samples. Nevertheless, partial inactivation was observed in three cases suggesting an inter-individual variability that could be due to fecal composition. The variability in feces composition could be related to differences in diet, gut microbiota or genetic background [60]. The identification of an intestinal viral reservoir in a subset of individuals with prolonged infection could be in agreement with the observed variability [55]. In addition, SARS-CoV-2 infection could be associated with an alteration of the gut microbiota composition, over the course of infection, that correlated with disease severity in Syrian hamster model [61]. Then, the gut microbiota diversity and physico-chemical parameters of feces could also explain the interindividual variability of SARS-CoV-2 inactivation in feces, as previously reviewed [5]. Comparing the viral persistence in stools from infected patients with divergent gastric symptoms may provide insight into the factors that may promote enteric replication of SARS-CoV-2.

## 5. Conclusions

We conclude from the present work that animal and human feces do not seem to constitute a favorable environment for the persistence of SARS-CoV-2 in an infectious form, contrary to enteric viruses such as Coxsackievirus. Although SARS-CoV-2 transmission after exposure to naturally contaminated feces is unlikely, we observed inter-individual variability, and we cannot rule out the possibility of persistent viral particles in human or animal feces, especially shortly after their excretion. To estimate the risk of contamination after exposure to contaminated feces, the infectious viral titer in the feces and the minimal infectious dose remain to be determined. These results must also be complemented with respect to the different circulating variants which have shown differences in pathogenicity, transmissibility, and also changes in tropism. If the mode of contamination in mink farms is probably related to the promiscuity imposed by the rearing conditions of these animals [62], the important carriage (35.8%) by free-ranging white-tailed deer in the USA suggests a mode of transmission from deer to deer that should be characterized [63]. If fecal-nasal contamination was involved in the spread of the virus in wildlife, this would potentially open up new pathways of the virus evolution.

## Figures and Tables

**Figure 1 viruses-14-01777-f001:**
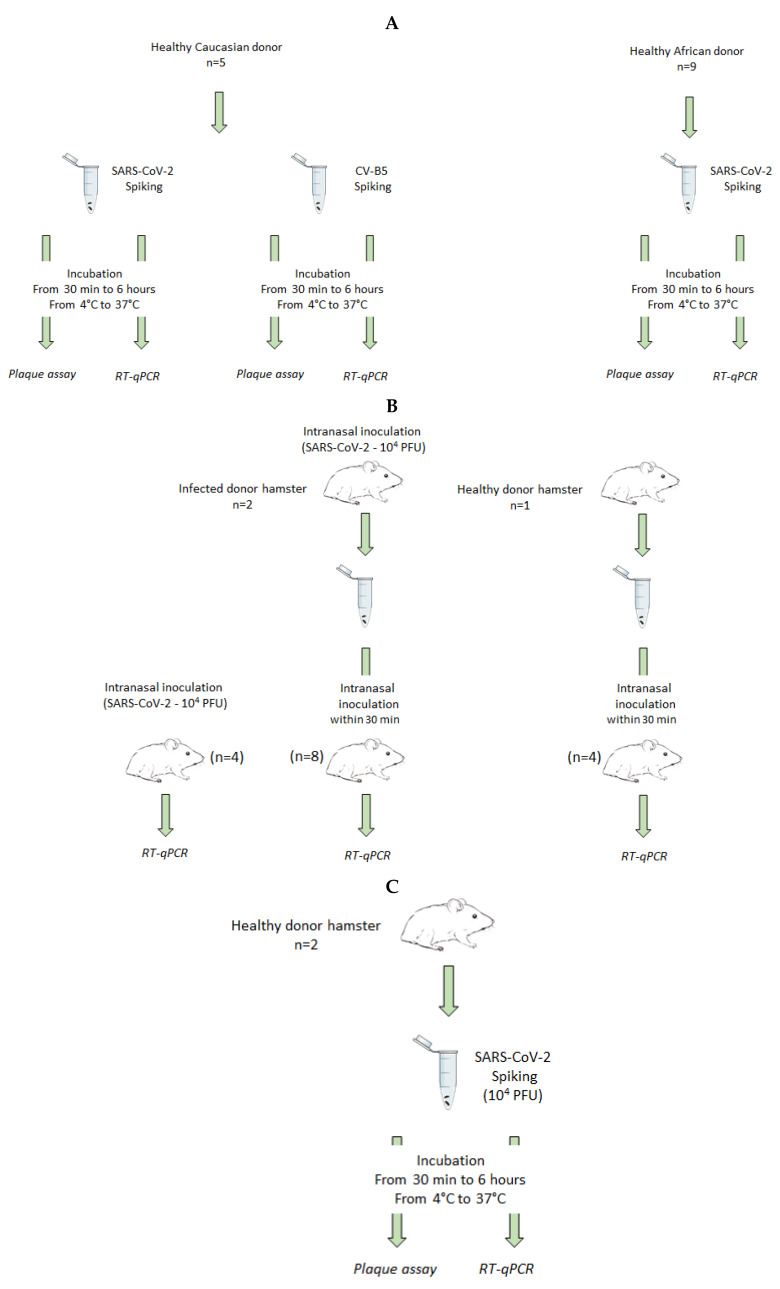
Experimental design used to evaluate the viral infection from contaminated (naturally or experimentally) feces. (**A**) Effect of incubation time and temperature on the persistence of viral genome and infectious virus in human stools from healthy donors by RT-qPCR and plaque assay, respectively. (**B**) Infection transmission after exposure to fresh feces from infected donor hamsters and persistence of the virus shedded in the feces by endpoint plaque assay and RT-qPCR. (**C**) Infection transmission after exposure to spiked fresh feces from healthy donor hamsters and effect of incubation time and temperature on the persistence of the virus spiked in fresh feces by endpoint plaque assay and RT-qPCR.

**Figure 3 viruses-14-01777-f003:**
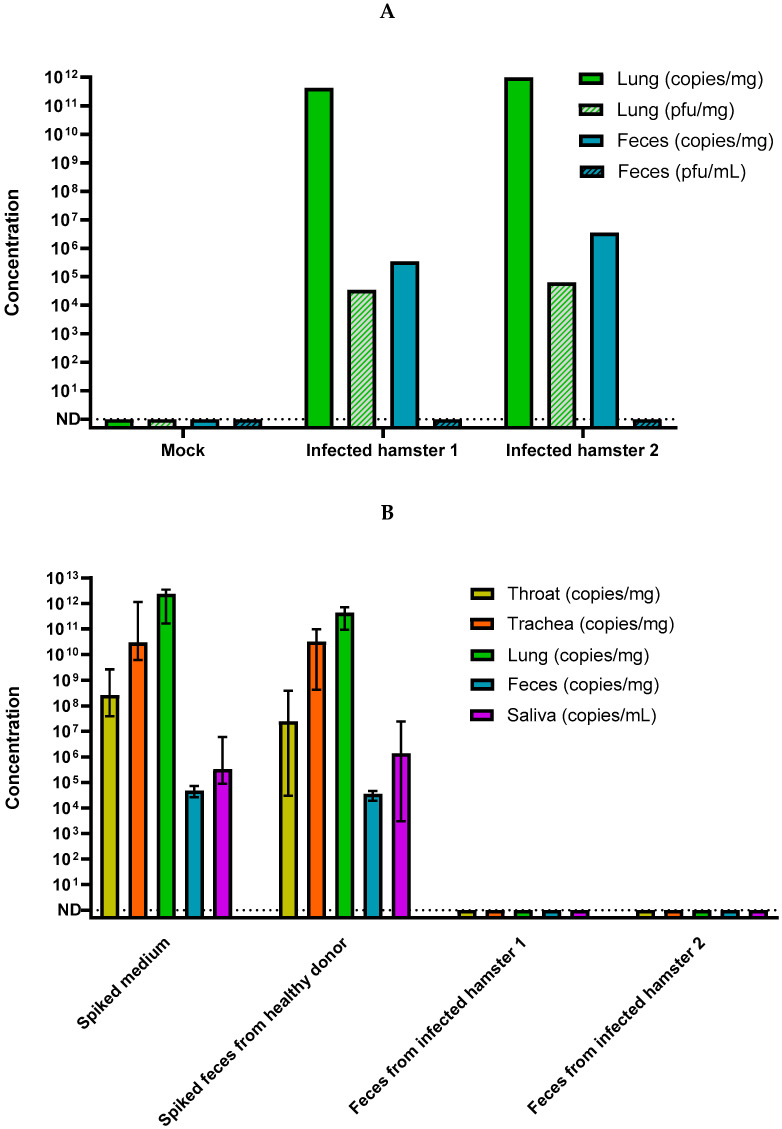
Detection of SARS-CoV-2 in different hamster organs 4 days post infection. (**A**) Quantification of viral RNA in SARS-CoV-2 infected donor hamsters by RT-qPCR and infectious particles by endpoint plaque assay was done in lungs and feces of animal. RNA genomes and infectious viruses were detected in lung whereas only viral RNA was detected in feces of infected animal. (**B**) Quantification of viral RNA in groups of recipient hamsters after intranasal instillation of fresh feces from infected donor specimens or feces from healthy donor spiked with infectious virus. As a positive control, hamsters were infected by virus spiked medium. Hamsters infected by virus-spiked feces samples developed similar infection pattern than specimen infected with positive control. Hamsters infected with feces from infected donor hamster did not present viral genomes in the different compartments analyzed. The concentration was measured by RT-qPCR and was indicated in copies per mg of tissue or mL of saliva.

**Figure 4 viruses-14-01777-f004:**
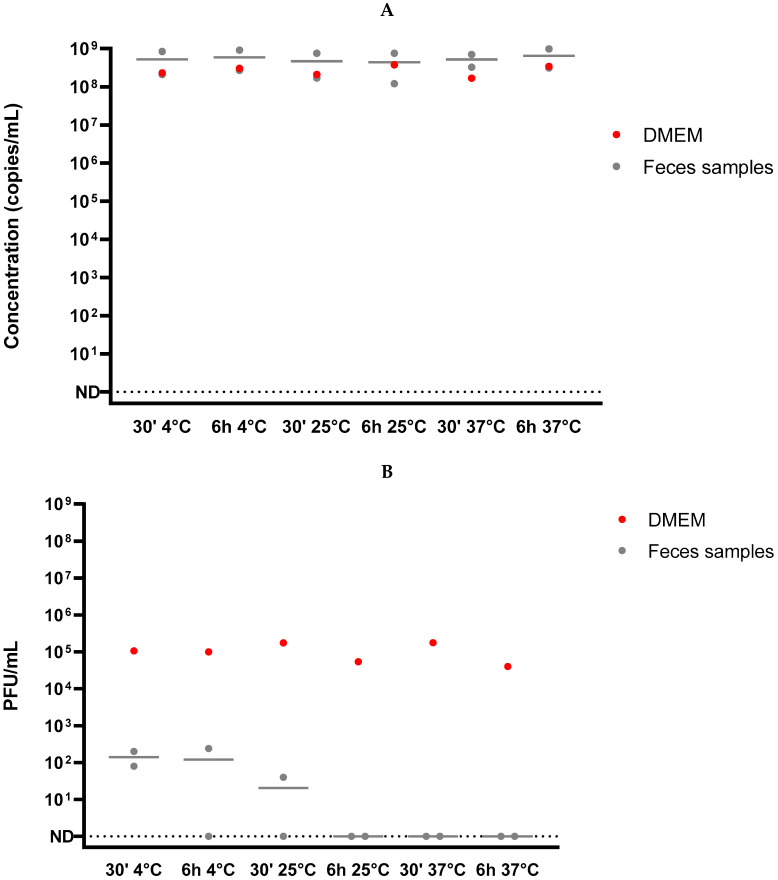
Persistence of SARS-CoV-2 in hamster feces at different incubation temperatures up to 6 h. (**A**) Viral RNA measured by RT-qPCR in hamster’s feces sample spiked with infectious virus (grey) was compared to virus incubated in cell culture medium (red) and did not show any change in RNA concentration after 6 h incubation at 4 °C or 37 °C. (**B**) Infectivity evaluation of SARS-CoV-2 particles by endpoint plaque assay after incubation in hamster’s feces (grey) and cell culture medium (red) was done. In these conditions, a decrease in infectivity was observed in time- and temperature-dependent manner.

## Data Availability

The authors confirm that the data supporting the findings of this study are available within the article and upon reasonable request.

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
