# Peer review of "Reduction in SARS-CoV-2 Virus Infectivity in Human and Hamster Feces"

_viruses, 2022, doi:10.3390/v14081777_

Round 1

Reviewer 1 Report

In this study, Wurtzer et al. used animal and human feces and concluded that infectious form SARS-CoV-2 persistence does not seems, contrary to enteric viruses like Coxsackievirus might be because of the differences in pathogenicity, transmissibility, and also changes in tropism. Moreover, if fecal-nasal contamination was involved in the spread of the virus in wildlife, this would potentially open up new pathways of the virus evolution.

Authors can find the comments and suggestions below

The manuscript seems to be interesting to the audience but it does need to revise the language to make it more understandable to the readers.

L66: gastrointestinal, if you once wrote a full name and then make an abbreviation then follow it thorough out the manuscript.

L6:  replace “lesions” with “viral lesions”

L88: Replace “it has been suggested” with “suggesting”

L123: Other studies have reported the higher prevalence and infectivity rate of SARS-CoV2 in men. why female hamsters were used in this study?

L186-191: All the human-taken samples were male or female or mixed. Mention properly.

I have seen various types of format variations/errors throughout the manuscript like font size, type, etc that show the non-serious attitude of the author as well as the journal’s initial quality checking team.

Author Response

Reviewer 1

We would like to thank for all your comments. We hope that we considered them correctly. Your comments are in bold in this document and our answer not. The line numbers are those of the corrected article with track changes.

-L66: gastrointestinal, if you once wrote a full name and then make an abbreviation then follow it thorough out the manuscript.

Replaced in lines 67 and 68.

-L67:  replace “lesions” with “viral lesions”

Done line 68.

-L88: Replace “it has been suggested” with “suggesting”

Done line 89.

-L123: Other studies have reported the higher prevalence and infectivity rate of SARS-CoV2 in men. why female hamsters were used in this study?

To our knowledge, male and female hamsters are good models of SARS-CoV-2 infection (Rosenke et al., 2022; Braxton et al., 2021). It is easier to handle females.

Rosenke K, Meade-White K, Letko M, Clancy C, Hansen F, Liu Y, Okumura A, Tang-Huau TL, Li R, Saturday G, Feldmann F, Scott D, Wang Z, Munster V, Jarvis MA, Feldmann H. Defining the Syrian hamster as a highly susceptible preclinical model for SARS-CoV-2 infection. Emerg Microbes Infect. 2020 Dec;9(1):2673-2684. doi: 10.1080/22221751.2020.1858177. PMID: 33251966; PMCID: PMC7782266.

Braxton AM, Creisher PS, Ruiz-Bedoya CA, Mulka KR, Dhakal S, Ordonez AA, Beck SE, Jain SK, Villano JS. Hamsters as a Model of Severe Acute Respiratory Syndrome Coronavirus-2. Comp Med. 2021 Oct 1;71(5):398-410. doi: 10.30802/AALAS-CM-21-000036. Epub 2021 Sep 29. PMID: 34588095; PMCID: PMC8594257.

-L186-191: All the human-taken samples were male or female or mixed. Mention properly.

Human samples were mixed. The text was completed lines 138 and 142.

-I have seen various types of format variations/errors throughout the manuscript like font size, type, etc that show the non-serious attitude of the author as well as the journal’s initial quality checking team.

We tried to harmonise font type in all the manuscript.

Reviewer 2 Report

Line 108 -113

Is there a order of importance or temporal in the aims? If you said that the hamster model was used to confirm human results, maybe you can change the order of aims at line 109 (human stools and hamster’s feces), and move lines from 166 to 226 at the beginning  of materials and methods (after line 121). If there isn’t a aims order, you can maintain current positions, but in the figure 1 human feces experiments are described in panel A and relative data obtained at the beginning of “results” paragraph.   

Lines 124 – 128

It could be better to indicate in the text what part of figure 1 is referred to hamsters inoculation (B and C I suppose)

Lines 133 - 136

Clarify the text respect to figure 1: In figure 1B and 1C are respectively described only two type of inocula: positive feces from infected donors and positive feces spiked with CoV2. In the text you are speaking about three inocula types (the third was negative sample from mock hamsters). Clarify also in the text “intranasal inoculation within 30 min” described only in figure 1B and 1C (in the results, page 11, description is present but there isn’t in materials and methods). Specify also  how the positive control was used.

Lines 140 – 141

I can’t understand corrispondence between text and figure 1 (part B and C); in the text you are speaking about “four group of 6 recipient hamsters” but in the figure sample sizes seem different (n = 4 n = 8 in B, and n = 4 n = 4 in C).

Line 145

Feces from non-infected hamsters: in figure 1C healty donor hamster n = 1, so please confirm if you used only one non-infected donor to obtain 2 feces pools

Line 145 – 146

You described two feces pools; please clarify how these pool were divided in different times and temperature conditions

Lines 160 – 161

“Sample were diluted serially…”, please specify dilution

Line 179

If it is possible, report Sars CoV2 variant

Line 191

Is there, also for african fecal samples, approval for human studies?

Line 202

I suggest you to use always the same name for endpoint plaque assay, please check in the text (for example line 202 endpoint dilution assay)

Line 205

Maybe it’s better to specify sample quantity

Line 228 – 231

Move these lines after paragraph “Titration” in materials and methods (after line 164)

Line 239 -249

Maybe it’s better to move details and phases of experiments in “materials and methods” section

Figura 2

I suggest you to specify in graphs temperature for 30’ (4°C)

Line 292

Figure 2 legend is too long because contain also some results; I suggest you to report results only in specific section

Line 295

Please correct “concentration” in “infectious concentration”

Line 332

Please clarify how did you obtained 13 PFU/animals

Line 354

“Viral RNa in hamster’s feces …(red)”; in the graph panels it seems the contrary

Author Response

Reviewer 2

We would like to thank for all your comments. We hope that we considered them correctly. Your comments are in bold in this document and our answer not. The line numbers are those of the corrected article with track changes.

-Line 108 -11: Is there a order of importance or temporal in the aims? If you said that the hamster model was used to confirm human results, maybe you can change the order of aims at line 109 (human stools and hamster’s feces), and move lines from 166 to 226 at the beginning of materials and methods (after line 121). If there isn’t a aims order, you can maintain current positions, but in the figure 1 human feces experiments are described in panel A and relative data obtained at the beginning of “results” paragraph.   

Done. The order was changed.

-Lines 124 – 128: It could be better to indicate in the text what part of figure 1 is referred to hamsters inoculation (B and C I suppose)

Done line 192. Figures 1B and 1C were changed too.

-Lines 133 – 136: Clarify the text respect to figure 1: In figure 1B and 1C are respectively described only two type of inocula: positive feces from infected donors and positive feces spiked with CoV2. In the text you are speaking about three inocula types (the third was negative sample from mock hamsters).

The text was correct in contrary of the figure1B. The figure was changed and we added “healthy mock” (line 202) in the text to harmonize text and figures.

 Clarify also in the text “intranasal inoculation within 30 min” described only in figure 1B and 1C (in the results, page 11, description is present but there isn’t in materials and methods).

We add a sentence to describe steps done in 30 min before inoculation on lines 203 to 206.   

Specify also how the positive control was used.

Specifications were added lines 206 to 208. “Positive control (104 PFU of SARS-CoV-2) were also be used to compare infection pattern (infected organs, clinical signs) between different inocula. It ensured the efficiency of the virus to cause an infection.”

-Lines 140 – 141: I can’t understand correspondence between text and figure 1 (part B and C); in the text you are speaking about “four group of 6 recipient hamsters” but in the figure sample sizes seem different (n = 4 n = 8 in B, and n = 4 n = 4 in C).

Text (line 211) and figure 1C and 1B were corrected.

-Line 145: Feces from non-infected hamsters: in figure 1C healty donor hamster n = 1, so please confirm if you used only one non-infected donor to obtain 2 feces pools

Text (line 216) and figure 1C were corrected. Two healthy hamsters were used.

-Line 145 – 146: You described two feces pools; please clarify how these pool were divided in different times and temperature conditions

Each pool from each hamster was tested in different times and temperature conditions. The text was modified line 221.

-Lines 160 – 161: “Sample were diluted serially…”, please specify dilution.

We specified the standard 10-fold dilution in text line 232.

-Line 179: If it is possible, report Sars CoV2 variant

It was a virus collected at the beginning of the pandemic. We add “collected in January 2020” (lines 130-131) to explain that.

-Line 191: Is there, also for african fecal samples, approval for human studies?

An ethical opinion of the national committee (Ethical and scientific opinion N° 00029/MSAS/DPRS/CNERS dated February 5, 2020) was obtained. The text was modified to be clear lines 143-144.

-Line 202: I suggest you to use always the same name for endpoint plaque assay, please check in the text (for example line 202 endpoint dilution assay)

Corrected lines 155, 231, 256, 258, 293, 303, 357 and 374.

Line 205: Maybe it’s better to specify sample quantity

Corrections (lines 158-159) was added in the text to explain what we did for each half of samples. 

-Line 228 – 231: Move these lines after paragraph “Titration” in materials and methods (after line 164)

Done in lines 236-240.

-Line 239 -249: Maybe it’s better to move details and phases of experiments in “materials and methods” section

We prefer to give again some details about time and temperature to explain our choice and to highlight results significance.

-Figure 2: I suggest you to specify in graphs temperature for 30’ (4°C)

Done.

Line 292: Figure 2 legend is too long because contain also some results; I suggest you to report results only in specific section

Results already described in the text was suppressed in figure 2 legend (lines 300 to 308).

-Line 295: Please correct “concentration” in “infectious concentration”

Sentence was suppressed (cf. previous comment).

-Line 332: Please clarify how did you obtained 13 PFU/animals

We used results of in vitro experimentation on hamsters feces (Figure 4B). The reduction of the infectious titer observed after 30 minutes at 4°C was about 2.9 log on figure 4B. To process the hamsters feces samples we need about 30 minutes and we worked at 4°C (centrifugation). With these data, we can calculate an approximate estimation of the dose inoculated to the hamster after processing the spiked feces.

Spiked feces with 104 PFU were processed. The new titer because of the reduction due to the process was about 10 4 -2.9 = 10 1.1 = 12.589 ≈ 13.

Text was modified on lines 337-338 to be more clear.

-Line 354: “Viral RNa in hamster’s feces …(red)”; in the graph panels it seems the contrary

Corrected from lines 371 to 375.

Reviewer 3 Report

The manuscript entitled “Reduction in SARS-CoV-2 virus infectivity in human and hamster feces” assess the infectivity of faeces derived from hamsters infected with SARS-CoV-2. Up to date there are few papers committed to this issue, and the evidence of stool infectivity is still scarce. The study is particularly interesting, well designed and properly performed. The conclusions are supported by obtained data, the discussion comprehensively describes the context of obtained results and indicates its novelty and importance. Language is very well, the whole paper is well written and it is a pleasure to read it, everything is clearly understandable. Obtained data are novel, and complementary to existing evidence. The subject is of particular importance for wide audience, highlighting that faeces from SARS-CoV-2 infected individuals present low risk of SARS-CoV-2 transmission, despite showing high load of viral genome. The study may be accepted for publication after checking the typographic errors (eg. line 35 rapidly, line 223 beta-actin). The Figures must be improved, since all of them are too big, thus I would suggest scale down the size of each chart to fit each whole figure on one page (for example figure 2: 2x3 smaller charts, clearly marked with A, B, C etc). The virus names should be included above each chart to make them easier to understand. Please change the position and increase the size (I would suggest to bold) of A, B, C signatures on the figures as now they are difficult to spot. The graduation between the logs on the chart scales is not necessary, the scale by 1 log will be sufficient.  

Author Response

Reviewer 3

We would like to thank for your comments. We hope that we considered them correctly. Your comments are in bold in this document and our answer not. The line numbers are those of the corrected article with track changes.

-The manuscript entitled “Reduction in SARS-CoV-2 virus infectivity in human and hamster feces” assess the infectivity of faeces derived from hamsters infected with SARS-CoV-2. Up to date there are few papers committed to this issue, and the evidence of stool infectivity is still scarce. The study is particularly interesting, well designed and properly performed. The conclusions are supported by obtained data, the discussion comprehensively describes the context of obtained results and indicates its novelty and importance. Language is very well, the whole paper is well written and it is a pleasure to read it, everything is clearly understandable. Obtained data are novel, and complementary to existing evidence. The subject is of particular importance for wide audience, highlighting that faeces from SARS-CoV-2 infected individuals present low risk of SARS-CoV-2 transmission, despite showing high load of viral genome.

Thank you very much.

-The study may be accepted for publication after checking the typographic errors (eg. line 35 rapidly, line 223 beta-actin).

Corrected.

-The Figures must be improved, since all of them are too big, thus I would suggest scale down the size of each chart to fit each whole figure on one page (for example figure 2: 2x3 smaller charts, clearly marked with A, B, C etc). The virus names should be included above each chart to make them easier to understand. Please change the position and increase the size (I would suggest to bold) of A, B, C signatures on the figures as now they are difficult to spot. The graduation between the logs on the chart scales is not necessary, the scale by 1 log will be sufficient.  

Done. Figures 1 (B and C), 2 (temperature of 4°C specified for 30’, virus names, size chart, graduation logs) and 4 (graduations logs, temperature of 4°C specified for 30’) were changed.
